:ᐧ️: PLOS | ONE

# Drone-based effective counting and ageing of hippopotamus (*Hippopotamus amphibius*) in the Okavango Delta in Botswana

**Victoria L. Inman** [1]*, **Richard T. Kingsford**[1], **Michael J. Chase**[2], **Keith E. A. Leggett**[1,3]

**1** Centre for Ecosystem Science, School of Biological, Earth and Environmental Sciences, UNSW Sydney, Sydney, NSW, Australia, **2** Elephants Without Borders, Kasane, Botswana, **3** Fowlers Gap Arid Zone Research Station, School of Biological, Earth and Environmental Sciences, UNSW Sydney, Sydney, NSW, Australia

* victoria.inman@outlook.com

**Data Availability Statement:** The minimal data set necessary to replicate the results is included as Supporting Information. The full data set with drone videos cannot be shared publicly in

## Abstract

Accurately estimating hippopotamus (*Hippopotamus amphibius*) numbers is difficult due to their aggressive nature, amphibious lifestyle, and habit of diving and surfacing. Traditionally, hippos are counted using aerial surveys and land/boat surveys. We compared estimates of numbers of hippos in a lagoon in the Okavango Delta, counted from land to counts from video taken from a DJI Phantom 4™ drone, testing for effectiveness at three heights (40 m, 80 m, and 120 m) and four times of day (early morning, late morning, early afternoon, and late afternoon). In addition, we determined effectiveness for differentiating age classes (juvenile, subadult, and adult), based on visual assessment and measurements from drone images, at different times and heights. Estimates in the pool averaged 9.18 (± 0.25SE, range 1–14, n = 112 counts). Drone counts at 40 m produced the highest counts of hippos, 10.6% higher than land counts and drone counts at 80 m, and 17.6% higher than drone counts at 120 m. Fewer hippos were counted in the early morning, when the hippos were active and most likely submerged, compared to all other times of day, when they tended to rest in shallow water with their bodies exposed. We were able to assign age classes to similar numbers of hippos from land counts and counts at 40 m, although land counts were better at identifying juveniles and subadults. Early morning was the least effective time to age hippos given their active behaviour, increasingly problematic with increasing height. Use of a relatively low-cost drone provided a rigorous and repeatable method for estimating numbers and ages of hippos, other than in the early morning, compared to land counts, considered the most accurate method of counting hippos.

## Introduction

Hippopotamus or hippo (*Hippopotamus amphibius*) are Vulnerable on the IUCN Red List of Threatened Species, numbering 115,000–130,000 [1]. Habitat loss and hunting for meat and ivory are driving declines [1], but much of the population data originates from aerial surveys,

accordance with the University of New South Wales' Animal Care and Ethics committee. Data are available from the UNSW's Animal Care and Ethics Committee (contact via animalethics@unsw.edu.au) for researchers who meet the criteria for access to confidential data.

**Funding:** VLI was supported by the Australian Government and the study by UNSW. The funders had no role in study design, data collection and analysis, decision to publish, or preparation of the manuscript.

**Competing interests:** The authors have declared that no competing interests exist.

which can be inaccurate [2–6]. Reliable and accurate spatial and temporal data on abundances and demographics of hippo populations are essential for effective conservation management [1,7,8] but hippos are inherently difficult to count because of their aquatic lifestyle and behaviour. They are also among the more dangerous animals in Africa [9–11], limiting effectiveness of on-land and water methods of counting [12].

Hippos are usually surveyed from the air [13–16], but also from boats and land [13,17,18]; each method has advantages and disadvantages. Aerial surveys cover large areas [4] but with limited time to scan waterbodies and count hippos, given their speed. Also, aircraft noise may cause hippos to submerge [19], contributing to underestimation [20]. Aerial surveys are costly and logistically difficult, resulting in long intervals between surveys [21–24]. Slow, low-flying microlight aircraft or helicopters capturing images may overcome some of these challenges [25] but remain costly and potentially logistically difficult, often still causing disturbance. Even counts from land which tend to be more accurate [4], given hippo pods can be observed for a long period of time, cannot identify the true number of hippos in a pod, even a small one, without marked or recognised individuals. This is because hippos in the water continually surface and submerge, and individuals are not easily distinguishable [12]. Accuracy improves when hippos rest in aggregations in shallow water ("rafting") [26] or on land, but still some individuals are inevitably obscured by others [27]. Although land counts are the most accurate method for counting hippos [2–4,28,29], they still have the potential to underestimate [30,31], and all counts should be considered a minimum, rather than a true count of hippos in an area [2,32,33]. Land counts can also be dangerous and difficult or impossible to do where hippo pods are in remote or not easily accessible areas [21,32]. Such difficulties compound when assessing demographic composition of hippo pods.

Drones (unmanned aerial systems/vehicles or remotely piloted aircraft) are an increasingly effective means for monitoring animals, including birds [34], turtles [35], dugongs [23], and cetaceans [36]. They usually have low impact, are relatively low cost, have consistent flight paths, allow remote operation away from wildlife, and enable monitoring of areas inaccessible by land or boat [23,35,37,38]. Hippos were counted, including their demographic composition, in pods in the Democratic Republic of Congo, using relatively expensive technology and sophisticated methods [39–41], but without comparing drone counts to a current survey method. Drone height and weather affected hippo detection, based on surveys only done in the early morning [41]. However, time of day is critical, given hippo behaviour varies throughout the day [3,20,42–45]. We trialled the use of a relatively low-cost drone, testing its effectiveness to estimate hippo numbers, the percentage of hippos that could be assigned to age classes, and numbers of juveniles, subadults, and adults, comparing these estimates to counts from land. We also tested how time of day and survey height affected these counts.

## Methods

We conducted land and drone counts over a lagoon (-19.41725˚E, 22.56815˚S, 2.4 ha) with a resident hippo population, within NG26 (Abu Concession) of the Okavango Delta, northern Botswana (Fig 1), over seven days (7th, 8th, 11th, 13th and 14th November 2017 and 2nd and 3rd December 2017).

### Drone surveys

The drone used was a multirotor DJI Phantom 4™ (1380 gram, 4K-quality video, 12.4 MP photo, aperture of f/2.8 [46]). The camera was controlled and stabilised by a three-axis gimbal, and the drone controlled by a GPS-stabilised system. All videos were filmed at 3840 x 2160 pixels (30 frames s$^{-1}$), with automatic ISO and shutter speed, allowing variation for neutrally

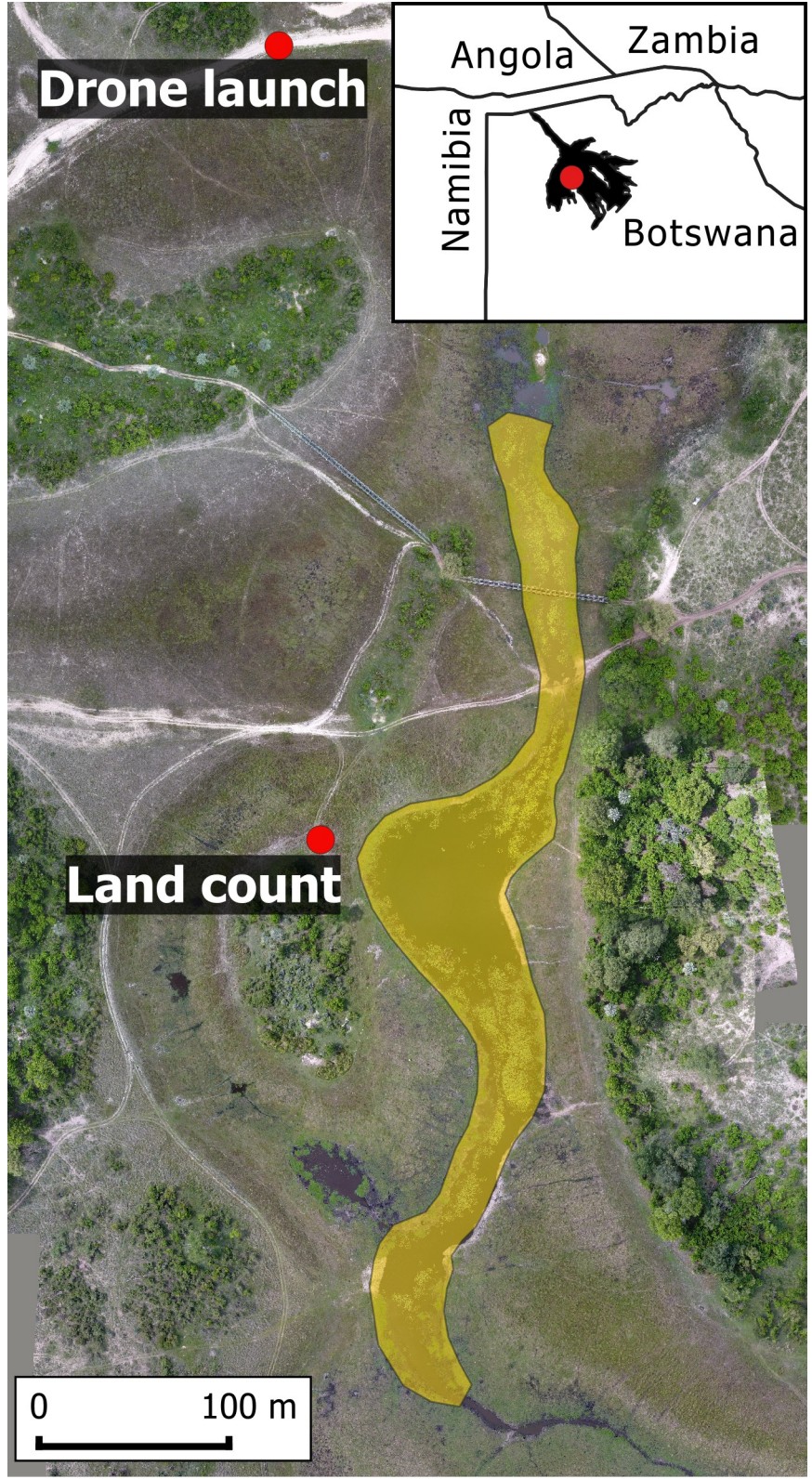

**Fig 1. Lagoon (shaded yellow) where hippos were counted in NG26, within the Okavango Delta in Botswana, showing where the drone was launched and land counts were done.** Note: background is an orthomosaic created using drone images.

exposed images. Sensor width was 6.2 mm and camera focal length was 3.61 mm [47]. The drone was programmed to fly (5.4 km hr$^{-1}$) in transects over the lagoon, calculated and controlled using the Drone Harmony app (www.droneharmony.com) and run through a smartphone, while continuously recording video. The lagoon was outlined using the satellite imagery provided on the app, and routes automatically calculated to ensure the entire surface area of the lagoon was captured on video, with a horizontal overlap of 20 percent, with the camera facing directly downward (gimbal angle of -90°).

We flew the drone at three heights sequentially, in descending order (120 m, 80 m, 40 m), starting with the height least likely to cause disturbance. The drone was launched and landed out of visual range of the hippos (Fig 1), avoiding disturbing them, only returning after two flights to change the battery. It took 30–40 minutes to complete the drone surveys. Given differential coverage of the lagoon, routes varied from backwards and forwards (east-west) across the lagoon (40 m, 80 m) to one path down the centre, north-south (120 m). Surveys were conducted four times a day (early morning 6:30–7:30 [EM]; late morning 10:00–11:00 [LM]; early afternoon 13:30–14:30 [EA]; late afternoon 17:00–18:00 [LA]), evenly dividing diurnal hours from an hour after sunrise to an hour before sunset, when there was maximum visibility. This resulted in twelve drone surveys per day (one at each height [120 m, 80 m, 40 m] at each time of day [EM, LM, EA, LA]), for seven days. Therefore, we completed a total of 84 drone counts (28 per height, 21 per time of day). For estimating numbers and measuring hippos, all 84 flight videos were randomised to avoid bias among proximate assessments. When reviewing the videos, we also checked for behaviours indicating the hippos were disturbed by the drone.

## Land surveys

We also counted hippos from a vehicle on land adjacent to the lagoon with two people (15 mins), in the same location each time (Fig 1) where all hippos could be observed, immediately following the last drone flight for each time of day. Therefore, we completed four land counts per day (EM, LM, EA, LA), for seven days, resulting in a total of 28 land counts. In conjunction with the drones counts, this equated to a total of 112 surveys. We noted any behaviours indicating hippos were disturbed by our presence, including submerging in water, vocalising, yawning, and charging [12,48].

## Measurement for ageing

Ground sampling distance (GSD, e.g. pixel size) was calculated based on the equation:

$$GSD = \frac{sensor\ width * flight\ height}{focal\ length * image\ width}$$

We used the 'snapshot' function of VLC media player [49] to obtain still images of each hippo visible on video. Individual images were imported into ImageJ [50], the 'set scale' function used to input the GSD for that image (1.79cm/pixel for drone images at 40 m, 3.58cm/pixel at 80 m, and 5.37cm/pixel at 120 m) and the 'straight line' function used to measure the length of each hippo from the tip of the snout to the base of the tail. This measured length was then used to assign each hippo to an age class, with no differentiation between males and females, based on the known relationship between body length and age [51]. Hippos < 184 cm and less than two years old were classed as juveniles (hippos produce a calf about once every two years [52]); hippos 184–233 cm long were two to four years old and classed as subadults, and hippos > 233 cm were classed as adults, given lower-end estimates of age of puberty in hippos is four years [51,53–55]. If the entire body was not visible (e.g. the hippo was partially submerged), but the visible section exceeded 233 cm, then it was classed as an adult. Other

partially submerged hippos, where the snout and base of the tail were not visible, were classed 'unknown'. For land counts, hippos were similarly assigned into the three age classes. Hippos judged as less than 1/2 the length of the largest hippo (typically the dominant male [56,57]) were classed as juveniles; subadults were between 1/2 and 2/3 the length of the largest hippo; and adults were over 2/3 the length of the largest hippo. Based on the proposed maximum hippo body length of 359cm [51], the land count classes aligned well with the drone classes, with the distinction between juveniles and subadults calculated as 179.50 cm (compared to 184 cm) and between subadults and adults as 239.30 cm (compared to 233 cm). During visual assessments, if hippos could not be assigned to an age class, they were recorded as unknown.

## Analysis

We tested the effect of survey height (including land counts), time of day, and their interaction on variations in total hippo count (model 1), percentage of hippos assigned to age classes (number of juveniles, subadults, and adults divided by the total count for each drone/land survey; model 2), and counts of juveniles, subadults, and adults (models 3–5). 'Height' had four levels (land count and drone heights 40 m, 80 m, 120 m) as did 'time of day' (early morning [EM], late morning [LM], early afternoon [EA], and late afternoon [LA]). Height and time of day were defined as fixed effects, with survey date as a random effect. Attempts to include a more complex random effects structure led to inadequate convergence in the models and so we adopted a simple random effect structure. For model 1, we used a linear mixed model, for model 2 a generalized linear mixed-effect model, with family Binomial and weights equal to the total number of hippos for each count, and for models 3–5 generalized linear mixed-effect models, with family Poisson, and a zero inflation variable. We checked for serial autocorrelation in the residuals of all models by comparing models with AR(1) covariance structures to models assuming uncorrelated residuals, using likelihood ratio tests (LRT, anova function). All modelling was conducted using the glmmTMB function (glmmTMB package [58]), with the significance of the fixed effects, their interaction, and the random effect, determined by comparing full and reduced models using LRT (anova function). Differences among the levels of the effects were tested using post hoc pairwise comparisons, based on estimated marginal means, using a Tukey adjustment with the emmeans package [59]. We determined the maximum number of hippos seen for each day (from any drone or land count), investigating how counts compared to this daily maximum, as another measure of accuracy, given that hippos generally do not move out of lagoons during diurnal hours [60]. The actual number of hippos in the lagoon on any particular day was unknown, and difficult to estimate given their behaviour, therefore true accuracy could not be calculated.

For all models, we examined plots of distributions of residuals against the predictors and Q–Q plots of the normal distribution to test the assumptions of homogeneity of variance and normality of data. These assumptions were met, requiring no transformation. All statistics were conducted using the R computing environment (version 3.5.2) [61]. Means were reported with standard errors.

## Ethics statement

This research was approved by UNSW's Animal Care & Ethics Committee (ACEC Number 17/75A), Civil Aviation Authority of Botswana (Remotely Piloted Aircraft Certificate Number RPA (H) 147) and The Republic of Botswana Ministry of Environment, Wildlife and Tourism (Research Permit EWT 8/36/4 XXXIII (55)).

## Results

The number of hippos counted in the lagoon averaged 9.18 ± 0.25 (range 1–14, n = 112 counts, S1 Table). We use daily maximum counts to describe the population because of potential emigration and immigration between surveys days: first two days (14 hippos), following two days (12 hippos), following two days (10 hippos), and last survey day (9 hippos). The pod consisted of one juvenile, two subadults, with adults ranging in number from eleven on the first survey day to six on the last day, based on daily maximum counts of each age class. All hippos remained in the water during the surveys. The drone's low impact sound was audible at 40 m (with decreasing noise level at higher altitudes) but hippos were not observed to be disturbed by the drone at any height, with no obvious changes in behaviour observed on the videos. The hippos were slightly disturbed by the presence of the vehicle during land counts. The hippos did not charge the vehicle or behave aggressively, but if they were near the edge of the pool when the observers arrived, they became vigilant and sometimes moved away from the observers. Their disturbance response varied with their activity, responding most when they were already active (e.g. the early morning), whereas if they were resting when we approached, they seldom moved.

Summary tables and post hoc comparisons of the fitted models are shown in Tables A-E in S1 Text. There was no significant interaction between height of survey and time of day on total hippo count ($\chi^2(9) = 11.276$, $p = 0.257$), so we omitted the interaction from subsequent analyses. Hippo count varied significantly with height ($\chi^2(3) = 12.180$, $p = 0.007$) and time of day ($\chi^2(3) = 38.384$, $p < 0.001$, Fig 2). Hippo count was also significantly negatively related to survey date ($\chi^2(1) = 64.757$, $p < 0.001$); fewer hippos were counted in subsequent days of the

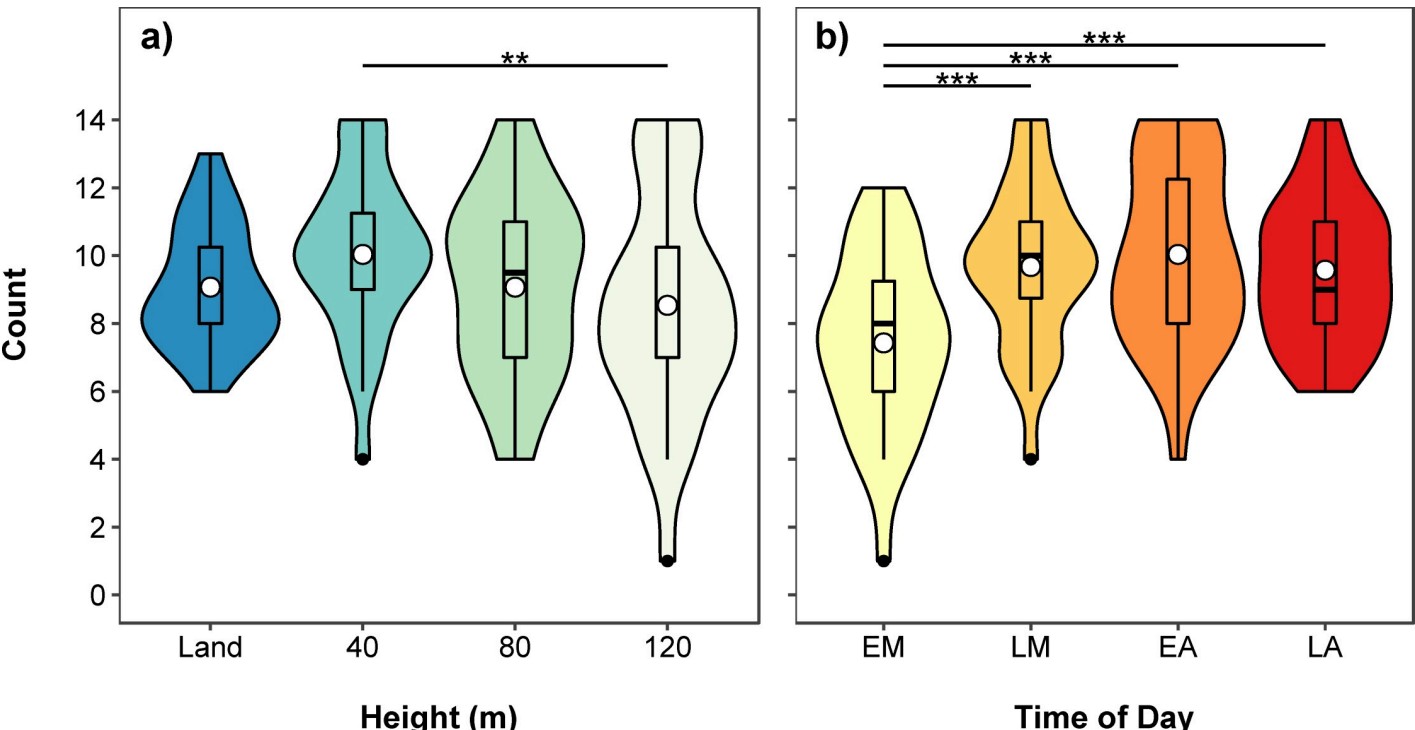

**Fig 2. Violin and boxplot (mean, circle) showing variation in total hippo counts for a) land and three drone heights (40m, 80m and 120m) and b) time of day (EM–early morning, LM–late morning, EA–early afternoon, LA–late afternoon).** Data collected using a Phantom 4 drone in a lagoon in the Okavango Delta in Botswana. Significant post hoc pairwise comparisons identified by asterisks.

survey. Counts at 40 m were significantly higher than at 120 m ($p = 0.004$), identifying on average 17.6% more hippos (Fig 2A). Also, 10.6% more hippos were counted at 40 m than during land counts, although this was not significant. The average number of hippos detected at 80 m was the same as the number of hippos counted from land, but numbers of hippos detected at 120 m were 5.9% less than during land counts. Early morning counts were significantly lower than at all other times of day (late morning, $p < 0.001$; early afternoon, $p < 0.001$; late afternoon, $p < 0.001$), with no significant differences among the other times of day (Fig 2B). There were 22.4–26.0% fewer hippos counted during early morning counts, compared to other times of the day. The inclusion of an AR(1) covariance structure did not improve model fit ($\chi^2(2) = 3.548$, $p = 0.170$), so was omitted from the model.

Our daily maximum counts occurred at all times of the day, although there were more in the middle of the day: early morning (3), late morning (6), early afternoon (9), and late afternoon (3, Fig 3, Table 1). Eighteen daily maximum counts were drone counts: 40 m (10), 80 m (3), and 120 m (5), along with three land counts (Fig 3, Table 1). The count with the greatest difference from the daily maximum was a 120 m drone count in the early morning (71.4% less hippos than daily maximum).

The percentage of hippos that were assigned to age classes was significantly related to the interaction between height and time of day ($\chi^2(9) = 17.100$, $p = 0.047$, Fig 4). Land counts and counts at 40 m assigned similar numbers of hippos to age classes and this did not differ with time of day. In the early morning, land counts assigned more hippos to age classes than counts at 80 m ($p = 0.013$) and 120 m ($p = 0.003$), with counts at 80 m and 120 m having significantly fewer hippos assigned to age classes in the early morning compared to the early and late afternoon (all $p < 0.05$). By late morning, all survey heights assigned similar numbers of hippos to age classes. The height and time of day survey with the highest average percentage of hippos assigned to age classes was land counts, in the late afternoon (66.8% of hippos), compared to the lowest average of 3.6% from surveys at 120 m in the early morning. The inclusion of an AR(1) covariance structure significantly improved model fit ($\chi^2(2) = 18.760$, $p < 0.001$), and was retained in the model.

There was no significant interaction between height and time of day on the number of observed juveniles ($\chi^2(9) = 7.994$, $p = 0.535$) or subadults ($\chi^2(9) = 12.640$, $p = 0.180$) and so we omitted the interaction from subsequent analyses. The number of juveniles and subadults was significantly related to height (juveniles, $\chi^2(3) = 19.172$, $p < 0.001$; subadults, $\chi^2(3) = 24.151$, $p < 0.001$, Fig 5A), with land counts providing significantly higher counts than counts at 40 m (subadults, $p = 0.021$), 80 m (juveniles, $p = 0.045$) and 120 m (juveniles, $p = 0.016$; subadults, $p = 0.007$). For juveniles, land counts were higher than counts at 40 m, although this was close to, but not below, the 0.05 significance level ($p = 0.072$). This was also true for subadults with land counts and counts at 80 m ($p = 0.055$). There were no significant differences among the other drone heights. The number of juveniles counted was not related to time of day ($\chi^2(3) = 3.158$, $p = 0.368$) but number of subadults was ($\chi^2(3) = 10.896$, $p = 0.012$, Fig 5B). Early morning counts of subadults were significantly lower than counts in the late afternoon ($p = 0.033$), with no significant differences among the other times of day. There was no effect of survey date on the number of juveniles or subadults. We were unable to include AR(1) covariance structures in the models due to lack of convergence.

The number of observed adults was significantly related to the interaction between height and time of day ($\chi^2(9) = 24.854$, $p = 0.003$, Fig 6). The number of adults counted from the land and at 40 m did not significantly change with time of day, but for the other heights, fewer adults were counted in the early morning compared to late morning (120 m, close to, but not, significant; $p = 0.067$), early afternoon (80 m, $p = 0.010$; 120 m, $p = 0.009$), and late afternoon (80 m, $p = 0.030$; 120 m, $p = 0.008$). From late morning onwards, and particularly in the

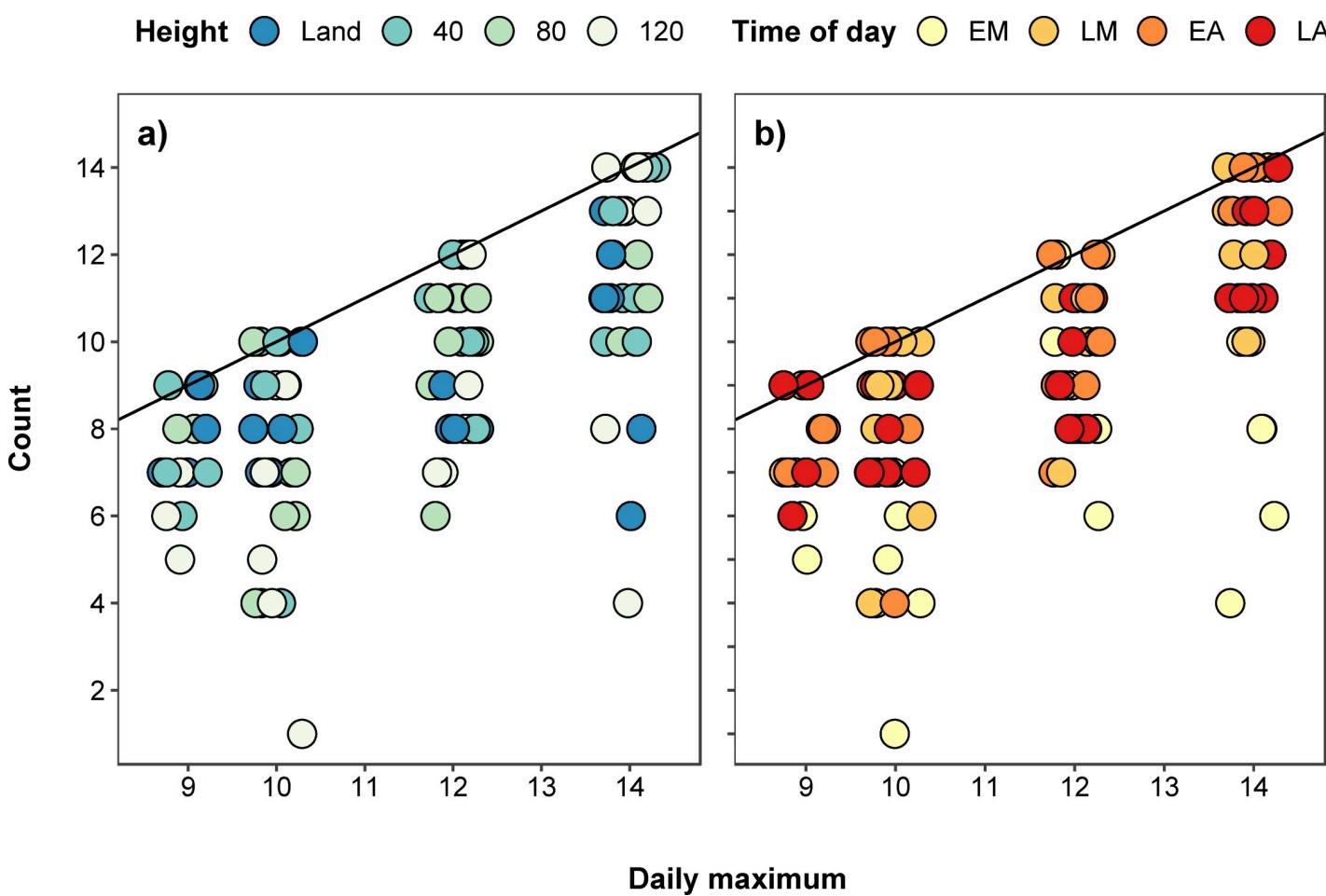

**Fig 3. Relationship between maximum hippo counts and a) different heights and b) times of day (EM–early morning, LM–late morning, EA–early afternoon, LA–late afternoon).** Points were jittered along the x axis.

afternoon, all surveys (land counts and drone counts at 40 m, 80 m, and 120 m) counted similar numbers of adults. There was no effect of survey date on the number of adults. The inclusion of an AR(1) covariance structure significantly improved model fit ($\chi^2(2) = 6.228$, $p = 0.044$).

## Discussion

We showed that by flying a relatively cheap drone at 40 m, as long as not in the early morning, reasonable estimates of hippo numbers and demographics were achievable. Although unable to estimate true accuracy for the drone counts, they were equal to or better than land counts, considered the most accurate method for counting hippos [2–4,28,29]. Both methods may still underestimate hippo numbers, which cannot be confirmed barring further testing where individuals can be recognised. Further, the models used may inadequately represent the random variation in the data, and results could be improved with larger sample sizes. The effectiveness of a reasonably low height reflects the increased video resolution with decreasing height; at higher heights it is difficult to distinguish and count individual hippos (Fig 2, S1 Fig). Land counts were possibly more accurate in our site than would be experienced in many other sites because the pod was reasonably habituated to humans and vehicles, given regular visits by

**Table 1. Mean counts of hippos (± SE) and number of times that each count matched the daily maximum for each observation time of day and height combination.** Sample size was seven for each count.

| Time of Day | Height | Mean count | Daily maximum[a] |
|---|---|---|---|
| Early Morning | Land | 7.3 ± 0.3 | 1 |
| | 40 m | 9.0 ± 1.1 | 2 |
| | 80 m | 7.7 ± 1.0 | 0 |
| | 120 m | 5.7 ± 1.1 | 0 |
| Late Morning | Land | 9.7 ± 0.7 | 0 |
| | 40 m | 10.7 ± 0.7 | 4 |
| | 80 m | 8.7 ± 1.2 | 0 |
| | 120 m | 9.6 ± 0.9 | 2 |
| Early Afternoon | Land | 9.7 ± 0.7 | 1 |
| | 40 m | 10.7 ± 1.0 | 4 |
| | 80 m | 10.3 ± 1.0 | 2 |
| | 120 m | 9.4 ± 1.4 | 2 |
| Late Afternoon | Land | 9.6 ± 0.5 | 1 |
| | 40 m | 9.7 ± 0.8 | 0 |
| | 80 m | 9.6 ± 0.8 | 1 |
| | 120 m | 9.4 ± 1.2 | 1 |

[a] The number of times this time of day/height combination was the count with the maximum number of hippos seen for that day.

tourists. Disturbance of less habituated hippos is likely in other places [33,62,63], leading to poorer land counts. Further, with increasing hippo pod size and water body size, land surveys become increasingly difficult, given uniformity of hippo appearance and diving behaviour [12].

Encouragingly, the drone did not disturb the hippos, although the pod's proximity to an airstrip (1 km) may have habituated the hippos to aerial noises. We surveyed a relatively small pod (maximum 14 individuals), whereas hippo pods can sometimes number in their hundreds [29,52]. A larger pod size would still be relatively easy to survey using a drone, though it would take longer to count and differentiate demographic groups; time-consuming data processing is a drone cost [24,64]. Increasingly, such data processing could lend itself to automation through machine learning, which has already proven successful at identifying hippos on thermal infra-red images [40], although it may be more difficult using RGB images, given the low colour contrast. Although hippos in larger congregations may be difficult to identify and track on video [41], our video continuously recorded the lagoon, allowing detection of hippos which surfaced momentarily, easily missed on images.

The lower height of the drone also allowed the demography of the hippo pod to be effectively estimated, with no significant difference in the percentage of hippos assigned to age classes between land counts and counts at 40 m (Fig 4). We identified more juveniles and subadults from our land counts, probably because they were easier to see than from drone footage and were able to be visually assigned to age classes, even when they were partially submerged. This was reflected in the similarity between drone and land counts of adults, given their relatively larger size, and because we classified partially submerged hippos over a certain size as adults on drone images. In addition, the small sample size of juveniles (one) and subadults (two) may have reduced the statistical power of our analyses. Estimating demographic groups in the Democratic Republic of Congo pods had mixed success, with numbers of each age class varying for different flights over the same pod [39]. This could reflect differential

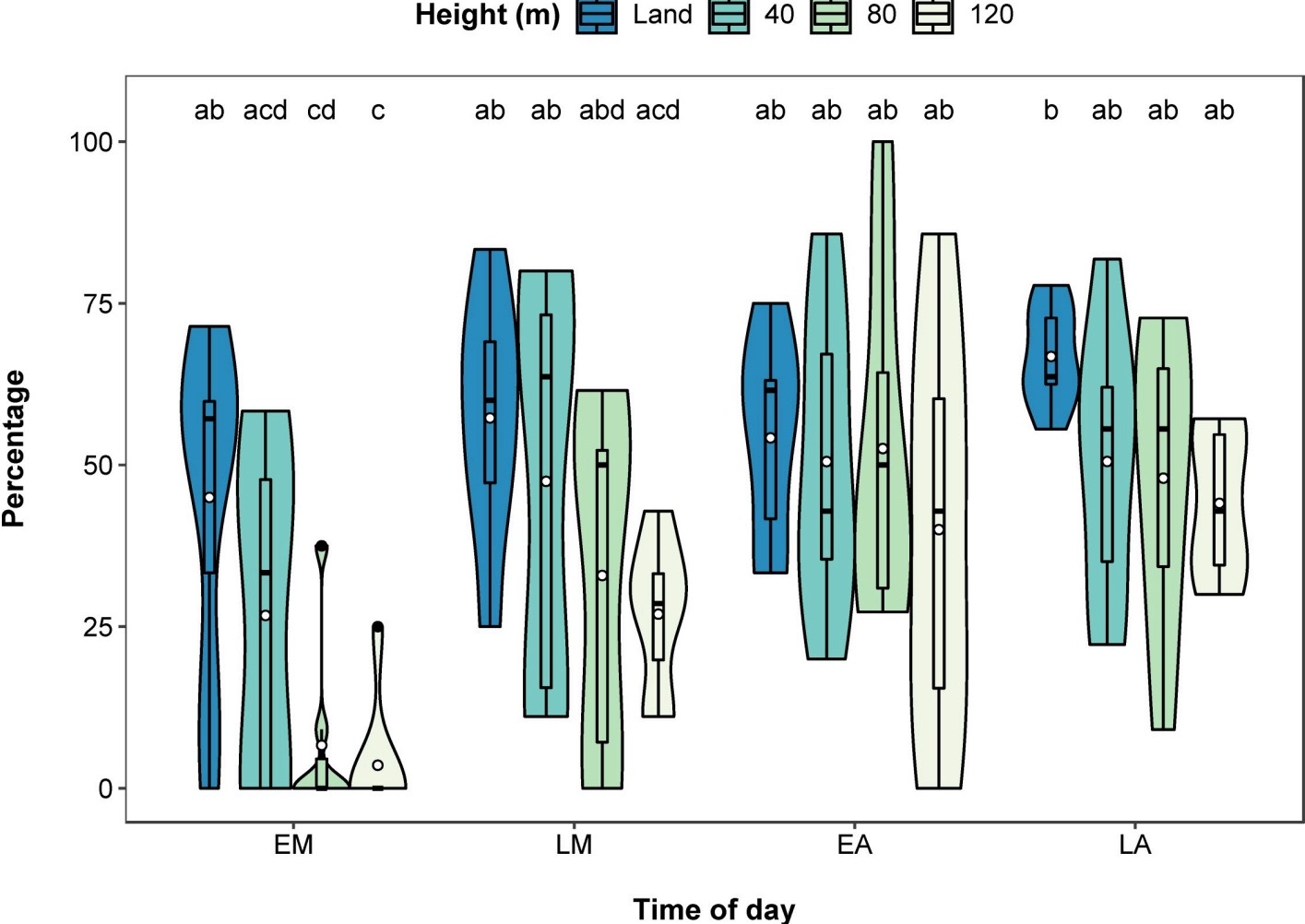

**Fig 4. Violin and boxplot (mean, circle) showing significant interactive effect of height and time of day (EM–early morning, LM–late morning, EA–early afternoon, LA–late afternoon) on percentage of hippos assigned to the three age classes (juvenile, subadult and adult).** Significant post hoc pairwise comparisons identified by letters.

hippo submergence between flights, unsuitable survey time of day, and visual extrapolation of body sizes. Restricting measurements to fully visible hippos reduces the sample size of hippos that can be assigned to age classes, but this can be increased by surveying when hippos are more exposed.

Aerial surveys of hippos are routinely flown at around 100 m at speeds of 160–180 km/hr [4,26,65–67], with observers estimating hippo numbers. It is unsurprising that these surveys underestimate hippos compared to land surveys [3,29,68]. There are clearly advantages to drone surveys; they capture hippo data at high resolution, given the relatively low flight height. The slower speed of the drone also increases viewing time, improving observations of hippos when diving and resurfacing. However, there are considerable advantages of the larger spatial coverage possible with aerial surveys, also accessing areas inaccessible for vehicles and drones. Inexpensive drones offer considerable promise for effective surveys of hippo pods, although battery life (about 20 minutes flight time) and flight range limits coverage to relatively small areas. The utility of drone technology is as an intermediate tool between lower accuracy, high cost, large scale aerial surveys and high accuracy but labour intensive land surveys [69].

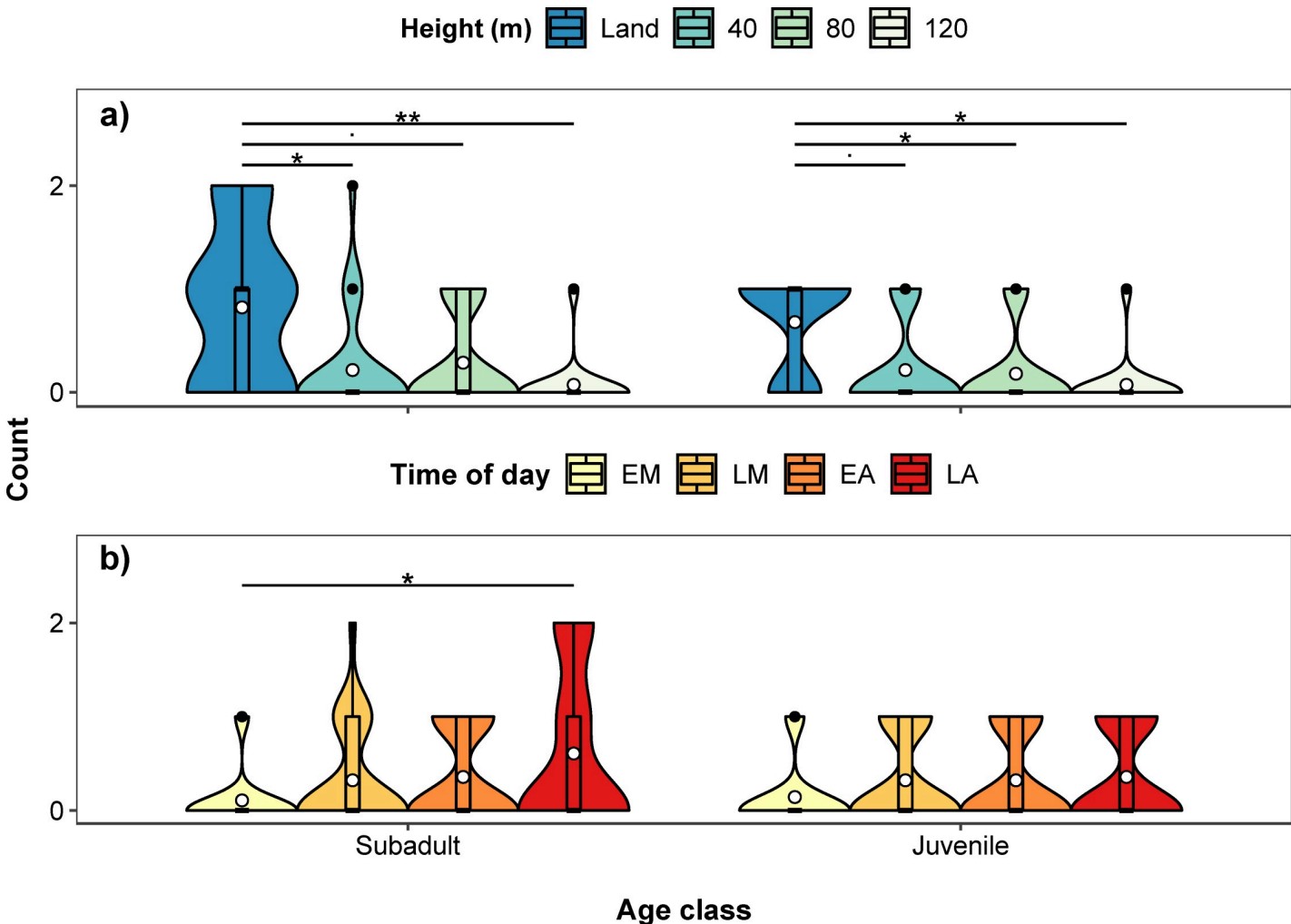

**Fig 5. Violin and boxplot (mean, circle) showing variation in counts for juvenile and subadult hippos for a) height and b) time of day (EM–early morning, LM–late morning, EA–early afternoon, LA–late afternoon).** Significant post hoc pairwise comparisons identified by asterisks, near significant (0.075 > p > 0.05) comparisons identified by dots.

Early morning was clearly an unsuitable time to effectively survey hippos (Figs 2–6). This is when hippos were very active, continuously diving and surfacing, after returning to the water from nocturnal feeding [20,43]. This could also be a problem late in the day, when there is high activity [13,43,48], but was not detected because our drone surveys occurred before sunset. Our highest hippo counts were in the late morning and afternoon when hippos usually rested as a group during the middle of the day [20,43], often in shallow water with most of their body exposed, making them easy to detect and distinguish (S1 Fig). Our avoidance of early mornings for hippo drone surveys runs counter to recommendations from surveys of Democratic Republic of Congo hippo pods [41], although these did not test the effect of time of day. Instead, they argued for the advantages of minimising sun reflection, which we effectively reduced by recording video, and surveying when they considered hippos most visible [27]. Hippo behaviour may differ by region or habitat, we therefore recommend adapting the timing of surveys to when hippo are resting, which may vary in location and time, and could be determined through simple observations, examining existing literature, or local knowledge where possible. Importantly, our surveys also effectively tracked changes in the hippo pod over

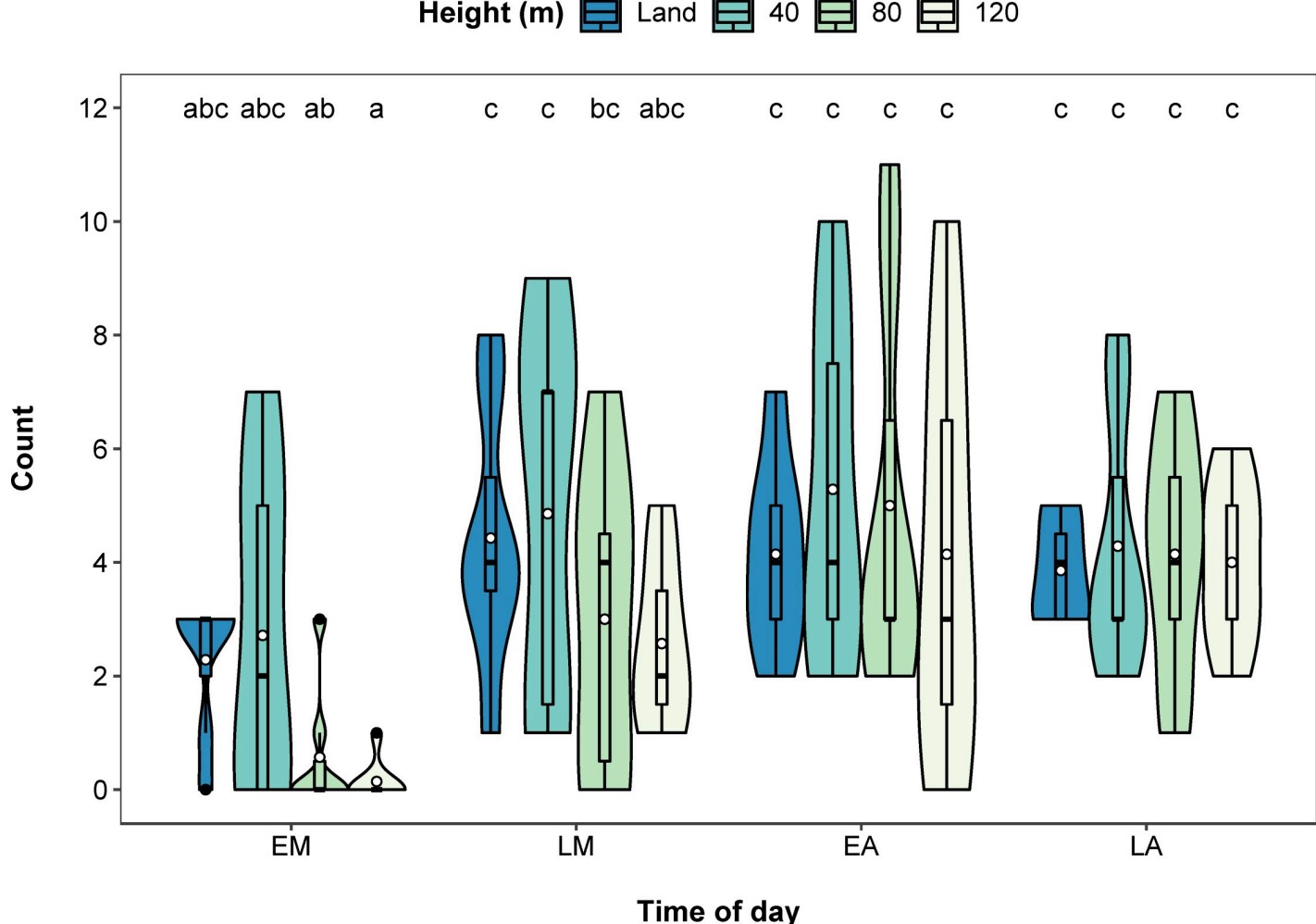

**Fig 6. Violin and boxplot (mean, circle) showing significant interactive effect of height and time of day (EM–early morning, LM–late morning, EA–early afternoon, LA–late afternoon) on counts for adult hippos.** Significant post hoc pairwise comparisons identified by letters. Note: the difference between counts at 120 m in the early morning and late morning was close to, but not, significant ($p = 0.067$).

time, as adults emigrated from the lagoon as it dried, a typical response of hippos to changing water availability [5,32].

Drones are increasingly valuable for monitoring wildlife populations [24,38,70], including hippos. Our analyses show that drone data can be used to estimate hippo pods, including their demographic structure. Importantly, they also provide a viable alternative to land based counts, with low impact on hippos, offering further opportunities to survey in difficult to access areas [3,29,32] and, just as critically, collect these data safely. Such data could be routinely collected in different river systems, providing indices of abundances, temporal changes and tracking the long-term status of hippo populations, an imperative given their declining populations in many parts of Africa.

## Supporting information

**S1 Table. Complete dataset.**
(CSV)

**S1 Text. Results of statistical models, as outputs from R.**
(DOCX)

**S1 Fig.  Snapshots of survey videos at a) 40 metres, b) 80 meters, c) 120 metres.** Notice the increasing difficulty of detecting hippos with increasing altitude due to lowering resolution. Images taken during early afternoon surveys showing resting posture of hippos with the majority of their body exposed, allowing easy detection and counting.
(PNG)

## Acknowledgments

We thank Elephants Without Borders for hosting the study and providing logistical support. In addition, we thank the Botswana Ministry of Environment, Wildlife and Tourism for affording us the opportunity to conduct this research. We thank Fly UAS for sponsoring Remote Pilot Licence training, Keboditse "CK" Mboroma for his assistance in the field, and Wilderness Safaris and staff at Abu and Seba Camps.

## Author Contributions

**Conceptualization:** Victoria L. Inman, Keith E. A. Leggett.

**Data curation:** Victoria L. Inman.

**Formal analysis:** Victoria L. Inman, Richard T. Kingsford, Keith E. A. Leggett.

**Funding acquisition:** Michael J. Chase, Keith E. A. Leggett.

**Investigation:** Victoria L. Inman.

**Methodology:** Victoria L. Inman, Keith E. A. Leggett.

**Project administration:** Victoria L. Inman, Keith E. A. Leggett.

**Resources:** Michael J. Chase, Keith E. A. Leggett.

**Software:** Victoria L. Inman.

**Supervision:** Richard T. Kingsford, Keith E. A. Leggett.

**Validation:** Victoria L. Inman, Richard T. Kingsford, Michael J. Chase, Keith E. A. Leggett.

**Visualization:** Victoria L. Inman.

**Writing – original draft:** Victoria L. Inman.

**Writing – review & editing:** Victoria L. Inman, Richard T. Kingsford, Michael J. Chase, Keith E. A. Leggett.

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
