## [Decision Letter · Decision Letter 0]

19 Sep 2019

PONE-D-19-18222

Drone-based effective counting and ageing of hippopotamus (Hippopotamus amphibius) in the Okavango Delta in Botswana

PLOS ONE

Dear Ms Inman,

Thank you for submitting your manuscript to PLOS ONE. After careful consideration, we feel that it has merit but does not fully meet PLOS ONE’s publication criteria as it currently stands. Therefore, we invite you to submit a revised version of the manuscript that addresses the points raised during the review process.

The main issues relate to the statistical analyses. Reviewer #2 provides some very specific suggestions concerning the models that should be used given the data you have on hand. Please address this issue with an appropriate re-analysis. Reviewer #2 also suggests that the degrees of freedom should be adjusted for possible bias due to small sample sizes. This should also be addressed.

Finally, is the "true" number of hippos actually known? If so, this should be used to assess the relative performance of the drone and land-based survey methods. I would imagine that for such a species, this might actually be possible!

On a personal note, you were very fortunate that the animals were not very bothered by the drone even at 40m. In my experience with horses and deer, animals appear to be very sensitive to the articifial buzz of a multirotor UAV. I wonder if perhaps the hippo's ear is not so sensitive to these sound frequencies (as compared to a horse, for example). 

We would appreciate receiving your revised manuscript by Nov 03 2019 11:59PM. To enhance the reproducibility of your results, we recommend that if applicable you deposit your laboratory protocols in protocols.io, where a protocol can be assigned its own identifier (DOI) such that it can be cited independently in the future. For instructions see: http://journals.plos.org/plosone/s/submission-guidelines#loc-laboratory-protocols

We look forward to receiving your revised manuscript.

Kind regards,

Tim A. Mousseau

Academic Editor

PLOS ONE

Journal Requirements:

2. We note that Figure 1 in your submission contains map/satellite images which may be copyrighted.

3. In your Methods section, please provide additional location information of the study area, including geographic coordinates for the data set if available.

5. Please amend your list of authors on the manuscript to ensure that each author is correctly linked to each affiliation. Authors’ affiliations should reflect the institution where the work was done (if authors moved subsequently, you can also list the new affiliation stating “current affiliation:….” as necessary).

Reviewers' comments:

Reviewer's Responses to Questions

**Comments to the Author**

1. Is the manuscript technically sound, and do the data support the conclusions?

Reviewer #1: Yes

Reviewer #2: Partly

2. Has the statistical analysis been performed appropriately and rigorously? 

Reviewer #1: Yes

Reviewer #2: No

3. Have the authors made all data underlying the findings in their manuscript fully available?

Reviewer #1: Yes

Reviewer #2: No

4. Is the manuscript presented in an intelligible fashion and written in standard English?

Reviewer #1: Yes

Reviewer #2: Yes

5. Review Comments to the Author

Reviewer #1: This is well executed research utilizing drone technology. As the technology develops to overcome the difficulties of drone control raised by authors in Discussion, this new techniques can be used widely in ecological monitoring.

Reviewer #2: My comments for the authors are contained in the attached document. The manuscript is fairly well written and easy to understand. The key weakness of the manuscript relates to the representation of the random variation in the data. As presented the models used do not adequately account for the potential total variation in the data.

6. PLOS authors have the option to publish the peer review history of their article (what does this mean?). If published, this will include your full peer review and any attached files.

Reviewer #1: No

Reviewer #2: No

---

## [Author Response · Author response to Decision Letter 0]

30 Oct 2019

Responses to specific reviewer and editor comments can be found in the "Response to Reviewers" pdf file attached to this resubmission.

---

## [Editor Report · Decision Letter 1]

21 Nov 2019

Drone-based effective counting and ageing of hippopotamus (Hippopotamus amphibius) in the Okavango Delta in Botswana

PONE-D-19-18222R1

Dear Dr. Inman,

We are pleased to inform you that your manuscript has been judged scientifically suitable for publication and will be formally accepted for publication once it complies with all outstanding technical requirements. Congratulations! 

With kind regards,

Tim A. Mousseau

Academic Editor

PLOS ONE

---

## [Editor Report · Acceptance letter]

27 Nov 2019

PONE-D-19-18222R1 

Drone-based effective counting and ageing of hippopotamus (*Hippopotamus amphibius*) in the Okavango Delta in Botswana 

Dear Dr. Inman:

I am pleased to inform you that your manuscript has been deemed suitable for publication in PLOS ONE. Congratulations! Your manuscript is now with our production department. 

With kind regards,

on behalf of

Dr. Tim A. Mousseau 

Academic Editor

PLOS ONE